# Effect of Moderate Exercise on the Superficial Zone of Articular Cartilage in Age-Related Osteoarthritis

**DOI:** 10.3390/diagnostics13203193

**Published:** 2023-10-12

**Authors:** Yukun Yin, Yuanyu Zhang, Li Guo, Pengcui Li, Dongming Wang, Lingan Huang, Xiaoqin Zhao, Gaige Wu, Lu Li, Xiaochun Wei

**Affiliations:** 1Shanxi Key Laboratory of Bone and Soft Tissue Injury Repair, Department of Orthopaedics, The Second Hospital of Shanxi Medical University, 382 Wuyi Road, Taiyuan 030001, China; yinyukun@sxmu.edu.cn (Y.Y.); zhangyuanyu@sxmu.edu.cn (Y.Z.); guoli@sxmu.edu.cn (L.G.); lipengcui@sxmu.edu.cn (P.L.); wangdongming@sxmu.edu.cn (D.W.); huanglingan@sxmu.edu.cn (L.H.); wugaige@sxmu.edu.cn (G.W.); 2Beijing Key Laboratory of Sports Injuries, Department of Sports Medicine, Peking University Third Hospital, Peking University, Beijing 100191, China; 3College of Physical Education, Taiyuan University of Technology, Taiyuan 030024, China; zhaoxiaoqin@tyut.edu.cn

**Keywords:** exercise, superficial zone, cartilage, degeneration, osteoarthritis

## Abstract

This study aimed to evaluate the effect of exercise on the superficial zone of the osteoarticular cartilage during osteoarthritis progression. Three-month-old, nine-month-old, and eighteen-month-old Sprague Dawley rats were randomly divided into two groups, moderate exercise and no exercise, for 10 weeks. Histological staining, immunostaining, and nanoindentation measurements were conducted to detect changes in the superficial zone. X-ray and micro-CT were quantitated to detect alterations in the microarchitecture of the tibial subchondral bone. Cells were extracted from the superficial zone of the cartilage under fluid-flow shear stress conditions to further verify changes in vitro. The number of cells and proteoglycan content in the superficial zone increased more in the exercise group than in the control group. Exercise can change the content and distribution of collagen types I and III in the superficial layer. In addition, TGFβ/pSmad2/3 and Prg4 expression levels increased under the intervention of exercise on the superficial zone. Exercise can improve the Young’s modulus of the cartilage and reduce the abnormal subchondral bone remodeling which occurs after superficial zone changes. Moderate exercise delays the degeneration of the articular cartilage by its effect on the superficial zone, and the TGFβ/pSmad2/3 signaling pathways and Prg4 play an important role.

## 1. Introduction

Osteoarthritis (OA) is the most prevalent chronic joint disease worldwide, and its incidence is expected to increase annually because of ageing and increasing obesity. For the non-surgical treatment of early- and mid-stage OA, lifestyle changes and physical activity are important alternatives to new effective drugs for long-term treatment; in addition, they may delay surgical intervention [1]. A combination of exercise regimens to reduce OA symptoms may be a potential treatment strategy [2,3].

Recently, the Osteoarthritis Research Society International (OARSI) proposed structured land-based exercise programs as the core treatment options for OA [4]. Moderate exercise is recommended as a non-drug intervention for OA [5]. Regular physical activity, especially aerobic exercises, can reduce pain and improve the function and health of patients with knee and hip OA [6,7,8]. Several animal studies have demonstrated the effects of exercise on arthritis symptoms. Planned exercise can reduce inflammation and pain in rodent models with chronic pain [9,10,11,12]. In addition, it can regulate the OA-related inflammatory response and protect the articular cartilage [13,14,15]. The pathogenesis of OA comprises many aspects, among which mechanical loads are common, suggesting that homeostasis of the articular cartilage may require appropriate mechanical loads, including shear stress in the superficial zone (SFZ) and compressive force in the deep zone. The integrity of the superficial zone, containing parallel collagen fibers and a relatively large number of flat chondrocytes, is essential for the protection and maintenance of the deep cartilage. This area is in contact with the synovial fluid and plays a crucial role in maintaining the integrity, elasticity, and other mechanical properties of the articular cartilage [16,17].

Cartilage degeneration is first observed with fibrosis in the SFZ; after the destruction of this zone, the cartilage in the deep zone degenerates [18]. There are two main types of collagens (types I and III) in the superficial zone of the cartilage. Fukui et al. reported that the expression of collagen type I was most enhanced in the SFZ of osteoarthritic cartilage [19]. Aigner et al. and Hosseininia et al. found deposits of type III collagen in the SFZ of osteoarthritic cartilage [20,21]. The chondrocytes of SFZ secrete numerous proteins that maintain their integrity and function. Superficial zone protein (SZP), also known as lubricin, is a secretory proteoglycan encoded by the Prg4 site and it is secreted by the superficial articular chondrocytes and synovial cells [22,23]. Lubricin and proteoglycans are essential for maintaining proper lubrication of the boundary of articular cartilage surfaces. There is growing evidence that the superficial zone plays various roles in joint homeostasis [24]. Therefore, we explored changes in the content and distribution of collagen and Prg4 in the superficial zone during exercise.

Under external mechanical stress, the SFZ of the articular cartilage is affected first (positively or negatively); afterward, the mechanical signal is transmitted to the middle and deep zone. Interestingly, mechanical stimulation can activate TGFβ signaling in chondrocytes [25,26]. Proteins in the TGFβ family can increase the expression level of SZP, stimulating the accumulation of SZP in superficial chondrocytes and synovial cells, and regulate the coefficient of friction in the joint [27,28]. In contrast, the inhibition of TGFβ/pSmad2/3 activity reduces the expression and accumulation of SZP and offsets the anabolic activity induced by mechanical stress [29]. Maintaining an appropriate level of TGFβ is essential for articular cartilage homeostasis and is closely related to mechanical stress.

Similar to mechanical loads of a certain degree, moderate exercise resolves inflammation and improves cartilage friction and lubrication properties. The direct protective effect of the SFZ in the joint cavity microenvironment is worth investigating. In the present study, we used moderate exercise methods [30] to explore the effects of exercise on the superficial zone of the osteoarticular cartilage.

## 2. Materials and Methods

Experimental animals

In this study, 3- (n = 16, body weight: 431.8 ± 16.9 g), 9- (n = 16, body weight: 519.7 ± 18.7 g), and 18-month-old (n = 16, body weight: 583.1 ± 24.5 g) male Sprague Dawley rats were purchased from Shulb Biotechnology Co., Ltd. (Wuhan, China). The rats were housed in individual cages with ad libitum access to food and water. The feed stuff was provided by SPF Biotechnology Co., Ltd. (Beijing, China) according to the national standards for rodent animal feed. The room temperature was 20 ± 3 °C, and the relative humidity was 55–70%. All procedures were approved by the Animal Ethics Committee of the Second Hospital of Shanxi Medical University (permit number: 2021007).

Animal models and treadmill exercise protocols

The 3- (n = 16), 9- (n = 16), and 18-month-old (n = 16) rats were randomly assigned to the following two groups (n = 8 each): control group and exercise group (treadmill exercise) [30]. Rats in the exercise group performed adaptive treadmill exercise for 1 week and thereafter maintained an appropriate intensity of exercise for 9 weeks (Monday–Friday, 45 min/day, treadmill speed of 20 m/min).

X-ray and micro-computed tomography (μCT) analysis

After 10 weeks of exercise or sedentary, anteroposterior and lateral radiographs of the right knee joint of 3- (n = 16), 9- (n = 16), and 18-month-old (n = 16) rats were captured to evaluate the development of osteophytes and joint degeneration using a small-animal X-ray apparatus (Faxitron UltraFocus USA). The joint space was exposed on anteroposterior radiography, and the knee joint was maintained at approximately 120° on lateral radiography, with the femoral condyles on both sides as close as possible. The exposure time and dose were set to ‘fully automatic’. The articular space width and patellar shape of the knee joint were evaluated based on the X-ray radiographs.

Tibial plateaus were harvested from 3- (n = 16), 9- (n = 16), and 18-month-old (n = 16) rats for in vitro μCT (μCT80; Scancomedical AG Switzerland) analysis. Scanning parameters were as follows: 70 kV, voxel size 15.6 μm, 0.5 mm aluminum filter, and exposure time, 250 ms. Ten cubic regions of interest (0.56  ×  1.5  × 1.5 mm^3^ each) were selected from the subchondral bone of the tibial plateau (Appendix A). The volume of interest (VOI) consisted of a stack of ROIs drawn over 36 cross-sections, resulting in a height of 0.56 mm. The VOI included the subchondral trabecular bone starting below the subchondral plate, and extending distally towards the growth plate, excluding both the cortical bone and growth plate interface. To quantitatively characterize the morphology and properties of the subchondral bone within the selected regions, bone volume fraction (BV/TV), trabecular number (Tb.N), trabecular thickness (Tb.Th), and trabecular separation (Tb.Sp) were calculated using manufacturer-provided 3D standard microstructure analysis (μCT Evaluation Program V6.6 SCANCO MEDICAL). All of the imaging was assessed by two experienced orthopedists; all disputes were resolved by discussion.

Nanoindentation measurements of cartilage tissue

After euthanasia, the tibial plateau of rats (n = 3 rats/group) was dissected free of tendon and ligament tissues and glued onto Optics11 (Holland) sample discs using a cyanoacrylate adhesive gel. After fixation, the samples were rehydrated in PBS for 15 min at RT. Before analysis, excess PBS was poured over the sample to prevent drying during measurement. A suitable probe (stiffness of 3–50 N/m) was selected for installation and immersed in a PBS solution for waveform detection; the appearance of a sinusoidal waveform indicated that the probe was fully usable. After inserting the flat hard surface of the probe in a liquid environment, the probe was pressed down to obtain the calibration factor, which is usually approximately 80% of the calibration factor in air. The probe was pressed on the cartilage surface in a liquid environment; the medial and lateral cartilages of each tibial plateau sample were assessed five times, and the average value was obtained for the final result (Appendix A). The measurements were fitted and calculated using the Hertz model, and the Young’s modulus of the cartilage was reported [31].

Histological staining and immunohistochemistry

Ten weeks after treatment, the rats (n = 5 rats/group) were sacrificed, and articular cartilage samples were collected. The knee joints were fixed in 4% paraformaldehyde for 24 h, decalcified in 10% EDTA (pH 7.4) for 6 weeks, and embedded in paraffin wax. The paraffin-embedded tissue was cut into 5 μm-thick sections. After sectioning and deparaffinization, the slides were stained using the Safranin O staining kit to examine the cartilage loss, according to the manufacturer’s instructions. For each section, positively stained cells were counted and the area occupied by the cells was measured in 20 fields. The degree of cartilage degeneration was assessed using the OARSI score [32]. Specifically, the number of cells arranged parallel to the articular surface and the area occupied by the cells were analyzed.

For immunohistochemistry, the sections were placed in 3% H_2_O_2_, incubated at room temperature in the dark for 10 min, and blocked at 37 °C with 10% goat serum for 30 min. Slices were then incubated overnight at 4 °C with primary antibodies COLⅠ (Bioss 1:100), COLⅢ (Bioss 1:100), Prg4 (Abmart 1:200), TGFβ (Abcam1:500), and pSmad2/3 (Abclonal 1:100), and the next day at room temperature for 40 min with secondary antibody. After sufficient washing, 3,3’-diaminobenzidine peroxidase substrate and hematoxylin solution were added. The IHC images were obtained using a Panoramic MIDI Scanner (3DHISTECH, Budapest, Hungary). Quantitative analysis was conducted in a blinded manner using Image-Pro Plus 6.0 software (Media Cybernetics, Rockville, MD, USA).

For Sirius Red staining, the slides were incubated with Sirius Red solution for 1 h. After washing with PBS to remove the residual staining solution from the surface, the nuclei were stained with a hematoxylin staining solution (Mayer type) for 3–5 min; afterwards, the slides were dehydrated frequently. Images were captured using a polarizing microscope. Type I collagen fibers were of intense orange-yellow or bright red color, while type III collagen fibers were green.

Isolation and culture of superficial zone cells

Superficial chondrocytes were isolated via transient enzymatic digestion as described previously [33]. Knee tissues were isolated from neonatal mice (P3–P5, n = 8), incubated with 0.25% trypsin (Gibco, Norristown, PA, USA) for 1 h, and digested with collagenase type 1 (Sigma SCR103, Darmstadt, Germany) for 1.5 h. Isolated cells were seeded on plates coated with 0.1% plasma fibronectin solution (Sigma F0895, Germany) for 2 h and thereafter blocked with 3% BSA for 30 min. After 20 min, unattached cells were washed twice with DMEM, and the attached cells were maintained in DMEM containing 10% fetal bovine serum. These cells were confirmed to be SFZ cells by detection of superficial zone and cell stemness markers (Appendix A).

Superficial zone cells under static or fluid-flow shear stress conditions

The SFZ cells (P1) were seeded in culture dishes in two groups (culture plate diameter = 10 cm). Cells of the control group were incubated in a static state at 37 °C. The cells of the fluid-flow shear stress (FFSS) group were incubated in an orbital shaker with a 20 mm radius of gyration (Sk-O180-S China) that was set to rotate at 2 rotations per sec. This exposed the chondrocytes to a stable laminar flow; the non-pulsating shear stress was approximately 7.6 dyn/cm^2^ [34]. After 8 h of incubation on a shaker, the cells were harvested for further experimentation [35].

Reverse-transcription quantitative polymerase chain reaction

SFZ cells (P1) were collected and total RNA was isolated using TRIzol™ reagent (Invitrogen). Total RNA concentrations were measured using NanoDrop Onec (Thermo Scientific, USA). Further, 1 µg total RNA was reverse transcribed into first-strand complementary DNA (cDNA) using PrimeScript™ RT Master Mix (Takara). The cDNA samples were used to conduct qPCR with an Applied Biosystems™ QuantStudio™ 6 Flex real-time PCR System in a 25 µL reaction using TB Green™ Premix Ex Taq™ II (Takara). Primer sequences are shown in Table 1.

Western blotting

Total proteins were extracted from the chondrocytes using radio immunoprecipitation assay lysis buffer, separated using 10% SDS-PAGE, and transferred to PVDF membranes. After blocking with 3% BSA, the membranes were incubated overnight at 4 °C with the following primary antibodies: Prg4 (Abmart 1:1000), TGFβ (Abcam 1:1000), and pSmad2/3 (Abclonal 1:1000). The membranes were subsequently incubated with secondary antibodies for 1 h at room temperature. Supersensitive ECL chemiluminescent substrates and an imager (BIO-RAD ChemiDoc XRS+ System, USA) were used, and the associated average optical densities of proteins were analyzed using ImageJ 1.4.3.67 software.

Statistical analysis

Statistical analyses were performed using SPSS statistical software (version 24.0). We performed unpaired Student’s *t*-test for in vivo and vitro statistics. Differences between groups were considered statistically significant at *p* * < 0.05.

## 3. Results

The changes in the articular cartilage in the early stage of OA were mainly concentrated in the superficial zone. Through exercise intervention in rats of different ages, we observed changes in the superficial articular cartilage and subchondral bone of the exercise group compared with those of the control group.

### 3.1. Subchondral Bone Change after Exercise

Radiographs and 3D reconstructions of the joints from the μCT analyses are shown in Figure 1. In the 3-month-old rat model, no significant changes were observed in the subchondral bone or joint space in radiography. An irregular bone surface, abnormal patellar shape, and narrower joint space were observed in the control group compared with the exercise group, including the 9- and 18-month-old rat models (Figure 1A,B). μCT analyses demonstrated that the BV/TV in the exercise group was significantly higher than that in the control group of 9- and 18-month-old rats’ medial compartment; Tb.N in the exercise group was higher than that in the control group in the 9-month-old rats’ medial compartment and the 18-month-old rats’ medial and lateral compartments; Tb.Th in the exercise group was higher than that in the control group in 9-month-old rats’ medial and lateral compartments and the 18-month-old rats’ medial compartment; and Tb.Sp in the exercise group was lower than that in the control group in the 3- and 18-month-old rats’ medial compartments (Figure 1B,C). These parameters indicated that exercise suppressed bone destruction in vivo.

### 3.2. Change in Cartilage Nanoindentation Modulus after Exercise

The prevention of the reduction in the Young’s modulus on the medial side markedly preceded that on the lateral side. The effect of exercise on the Young’s modulus of the medial side was observed as early as in the 3-month-old rats and continued in the 9- and 18-month-old rats, whereas a significant prevention of reduction in the Young’s modulus of the lateral space was observed only in 18-month-old rats (Figure 2A,B).

### 3.3. Change in the Number and Distribution of SFZ Chondrocytes after Exercise

Safranin O/Fast Green staining was used to visualize the severity of cartilage damage in the knees (Figure 3A), and articular cartilage damage was quantified using OARSI scoring. The OARSI scores of the control group were markedly higher than those of the exercise group of 9- and 18-month-old rats, whereas there were no significant differences between the exercise and control groups of 3-month-old rats. Interestingly, we obtained different results in the SFZ of the cartilage. At 3 months of age, the numbers of cells parallel to the surface of the cartilage and the Safranin O-stained area of the SFZ (yellow arrow) in the exercise group were significantly higher than those in the control group (Figure 3B).

### 3.4. Change in the Distribution and Content of Type Ⅰ and Type Ⅲ Collagens in SFZ after Exercise

Type I and type III collagens were detected by immunohistochemistry. The expression of type I collagen decreased, whereas that of type III collagen increased, in the exercise group compared with the control group in 3- and 9-month-old rats (Figure 4A,B). These results were also obtained in Sirius Red staining. The staining intensity of type I collagen (intense orange-yellow) in the exercise group was significantly higher than that in control group of 3-month-old rats, whereas the staining intensity of type Ⅲ collagen (green) in the exercise group was lower than that in control group of 3-, 9-, and 18-month-old rats (Figure 4C). In addition, exercise mainly delayed the degeneration of SFZ collagen fibers by changing the type I: type III collagen ratio.

### 3.5. Change in Expression of Lubricin/Prg4 and TGFβ-pSmad2/3 in SFZ after Exercise In Vivo

Owing to the close relationship between the superficial zone of the cartilage and Prg4, we examined the expression of Prg4 in the exercise and control groups of rats at different ages by using immunohistochemistry. We determined that moderate exercise increased Prg4 expression in the SFZ of cartilage, which was confirmed in the 3-, 9-, and 18-month-old rats (Figure 5A). Moreover, TGFβ-pSmad2/3, as a classical pathway of cartilage mechanical conduction, is closely related to exercise and mechanical load. TGFβ-pSmad2/3 expression increased in the exercise group compared with that in the control group of 3-, 9-, and 18-month-old rats, a result that was consistent with the trend of increasing Prg4 expression (Figure 5B,C).

### 3.6. SFZ Cells Change after FFSS In Vitro

Harvested SFZ cells under static or FFSS conditions are shown in Figure 6A. We detected the expression of these key factors in the harvested SFZ cells under static or FFSS conditions (Figure 6A). After exercise intervention, there was a significant increase in the expressions of TGFβ, Smad2, and Prg4 at the gene level, whereas there was no change in Smad3 expression (Figure 6C). TGFβ, pSmad2/3, and Prg4 expressions increased at the protein level (Figure 6B,D).

## 4. Discussion

In the present study, we evaluated changes in SFZ development and the expression of associated proteins and molecules in a rat treadmill exercise model over time. Our results showed that moderate exercise can delay cartilage degeneration in the early stages of knee osteoarthritis by its effects on the SFZ of the cartilage and subchondral bone. Changes in the SFZ of the cartilage were the primary focus of this study. We used a spontaneous OA rat model instead of the conventional post-traumatic OA model because it is more consistent with the natural course of OA development and allows the observation of changes in the SFZ in a more specific, detailed, and realistic way. We did not use older rats in this study for two reasons. Firstly, the main purpose of our study was to investigate the changes in the superficial zone of articular cartilage, which is completely worn out in the knee joints of older rats and cannot be observed. Secondly, older rats may have a reduced tolerance to the moderate amount of exercise used in this study, which may lead to negative effects. At present, the clinic shows that it is older patients who suffer from osteoarthritis. What may explain this result is that most of the research on the pathogenesis and treatment of cartilage degeneration mainly focuses on the overall process of OA, ignoring research on the early stage of cartilage degeneration. Most patients are not diagnosed with OA until they develop relevant symptoms. And the reason why we focused on the superficial zone of cartilage was that we found that, in the very early stages of degeneration, there was no change in the middle and deep zone of the articular cartilage while the superficial zone of the articular cartilage was significantly changed in the histological diagnosis. Our findings that moderate exercise intervention in middle and old age can delay the degeneration of the articular cartilage by protecting the environment of the superficial cells and cell matrix of the articular cartilage may provide a new idea for the prevention of early cartilage degeneration at a relatively young age rather than in old age.

Standard mild or moderate exercises had a beneficial effect on the severity of cartilage lesions in a rat model with OA [30,36]. Moderate physical activity and normal mechanical joint load in older rats can improve the tribology and lubrication properties of the articular cartilage, thereby preventing cartilage degradation [37]. In addition, there are many benefits for patients with knee OA. These benefits are sufficient to stimulate interest in the exploration of related mechanisms. Several studies have shown that physiological exercise loading suppresses osteoarthritis progression through its effect on the thickness and glycosaminoglycan (GAG) content of the articular cartilage, subchondral bone, synovium, and microenvironment of the cartilage. However, there is limited research on the effect of exercise in the SFZ and its regulatory mechanisms because of the lack of knowledge on the function of the SFZ. Additionally, it is more difficult to observe the SFZ of the cartilage than the whole cartilage tissue.

The SFZ has a unique structure with a low proteoglycan content, high density of flat cells, and collagen orientation parallel to the joint surface. The surface is well lubricated, with a low coefficient of friction [38]. Because of these characteristics, the functions and effects of the SFZ differ from those of the deep cartilage tissue and cells. In the present study, the number of unique flat cells in the SFZ and the surrounding matrix GAG increased after exercise. This was also demonstrated by an increase in cell viability under FFSS conditions in vitro. These changes are essential for maintaining the integrity of the SFZ. Furthermore, these changes were observed as early as in the 3-month-old rats. In the present study, the focus on the SFZ compensated for the absence of changes noted in previous studies on early-stage OA where the entire cartilage tissue may not change significantly.

Lubricin/Prg4, a relatively well-studied functional protein of the superficial surface of articular cartilage at this stage, is mainly distributed on the superficial surface. It maintains joint surface lubrication [24], acts as a signaling molecule [39,40,41], and regulates the differentiation of SFZ cells [42]. Among the many signaling pathways that can regulate Prg4 expression, we focused on the TGFβ-pSmad2/3 signaling pathway. The TGFβ-pSmad2/3 pathway is a classical pathway that affects cartilage mechanical conduction and is closely related to the mechanical load we studied in this experiment [43,44]. However, TGFβ plays an important role in inducing Prg4 expression in muscle-derived mesenchymal stem [45] and synovial cells [46]. Therefore, we investigated the effect of the TGFβ pathway on Prg4 expression in superficial cartilage cells with exercise intervention. Appropriate mechanical stress stimulation may transform mechanical signals into biological signals through the TGFβ pathway and act on Prg4, thereby affecting the function and morphology of the SFZ of articular cartilage. To further verify changes in the TGFβ pathway and Prg4 expression and their regulatory mechanisms in vitro, we used SFZ cells under the FFSS condition to simulate changes in the superficial cartilage in response to shear stress, which is the main force acting on the SFZ in vivo. In addition, we demonstrated that TGFβ, pSmad2/3, and Prg4 expressions increased synchronously under shear force. These findings may lead to novel therapeutic strategies for early-stage OA and further reveal the mechanisms underlying joint homeostasis.

Collagen is the most abundant protein of the ECM protein family, accounting for two-thirds of soft shaft mass in adult joints [47]. Type II collagen and aggregation glue are the main ECM proteins in the cartilage. Although only a small fraction of the mature matrix is present, these secondary collagens not only play important structural roles in the mechanical properties, organization, and shape of the articular cartilage but also perform specific biological functions [48]. The progressive degeneration of the cartilage involves the degradation of matrix components, including secondary collagens.

As shown by our experimental results, we focused on type I and III collagens in the SFZ. In the SFZ of normal and early-stage OA cartilage, type I collagen content was higher than that in advanced- and late-stage OA cartilage, and its expression was further enhanced by exercise intervention. Therefore, when exposed to mechanical stress, type I collagen is likely synthesized as a modifier of the existing fibril network in response to tissue and matrix damage. Type I collagen content gradually decreased with increasing age, which was inconsistent with the changes observed in the conventional PTOA model. It was considered that the mechanism of joint metabolic imbalance and significant loss of chondrocytes with age were different from those of young OA caused by the trauma model. The age-related decrease in collagen outweighed the compensatory increase caused by traumatic injury owing to the limited repair capacity of the articular cartilage in the elderly. This is one of the major reasons we used a spontaneous OA model in the present study. Furthermore, this suggests that, compared with type III collagen, type I collagen may have a more direct and important relationship with age. The filamentous polymer of type III collagen may add cohesion to a swollen and perhaps weakened existing collagen II fiber network [49]. Notably, the earliest changes observed in the articular cartilage in experimental animal models of OA were swelling of the collagen fiber network. A reduction in collagen III leads to thickened collagen fibers and increased collagen cross-links [50]. Type III collagen expression is part of the early response repertoire of articular chondrocytes on intact articular surfaces of joints undergoing progressive osteoarthritic molecular failure. Although this expression occurs in normal adult joints, it is significantly more pronounced in the OA cartilage [21]. We found that exercise increased type I collagen expression and decreased type III collagen expression in the SFZ at an early stage when the whole cartilage appeared macroscopically intact. However, the collagen network was altered in the SFZ due to collagen fiber thinning, in addition to the change in collagen types. This phenotypic alteration of collagen may result from the production of mechanically dysfunctional ECM; in addition, it is thought to be an important mechanism leading to the loss of cartilage homeostasis in OA.

The temporal sequences of changes in the superficial and subchondral bone zone were examined. Nanoindentation experiments were performed to examine the changes in the biomechanical properties of the cartilage caused by early changes in the SFZ. μCT is mainly used to detect changes in the trabecular bone in the subchondral bone. Some findings suggest that subchondral bone changes might precede cartilage degeneration during OA, without considering the SFZ of the articular cartilage [51]. Combined with the previous results, the comparison between the superficial and subchondral bones revealed that exercise had no significant effect on the trabecular bone mass, thickness, and separation in the subchondral bone of 3-month-old rats, whereas the biomechanics, cell number, and surrounding cellular matrix of the superficial cartilage changed significantly under exercise intervention. In both 9- and 18-month-old rats, the biomechanical changes in the cartilage affected by exercise were consistent with the changes in the subchondral bone, both of which indicated that the influence of exercise on the medial tibial plateau of the knee joint was earlier and greater than that on the lateral plateau. The effect of exercise on the SFZ occurred earlier than on the subchondral bone, and the effect on the medial plateau occurred earlier than on the lateral plateau.

## 5. Conclusions

Our study demonstrated that exercise played a significant role in delaying the degeneration of the superficial zone of the knee articular cartilage during age-related osteoarthritis progression through the TGFβ-pSmad2/3 pathway, which converts mechanical signals of mechanical loading into biological signals and affects Prg4 expression. These results provide a new insight into the function and structure of the SFZ and the influence of nonoperative therapy on early-stage OA.

## Figures and Tables

**Figure 1 diagnostics-13-03193-f001:**
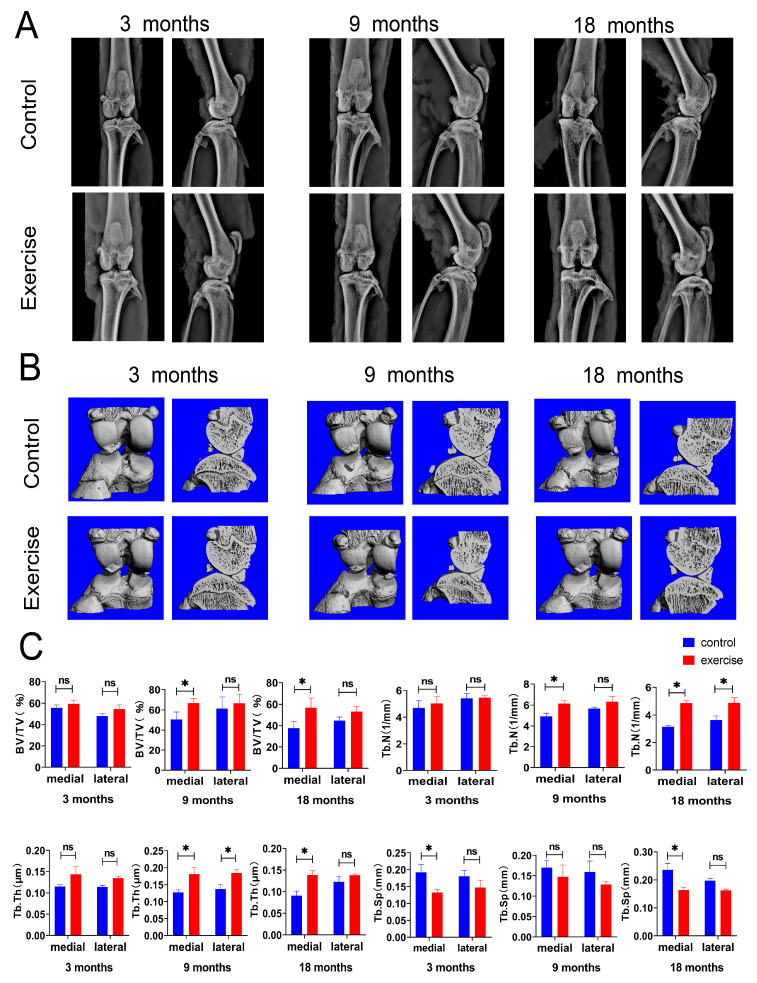
Exercise attenuated knee degeneration and enhanced the subchondral bone mass of rats. (**A**) Anterior and lateral knee X-rays of rats aged 3 months, 9 months, and 18 months (n = 8 each) in the exercise and control groups. (**B**) Representative three-dimensional micro-CT images of sagittal and coronal views of subchondral bone of 3-, 9-, and 18- month-old rats (n = 8 each) in the exercise and control groups. (**C**) Analyses of BV/TV, Tb.N, Tb.Th, and Tb.Sp are shown. Data are represented as mean ± S.E.M. *p* * < 0.05. ns: no significance. BV/TV: bone volume fraction, Tb.N: trabecular number, Tb.Th: trabecular thickness, Tb.Sp: trabecular separation.

**Figure 2 diagnostics-13-03193-f002:**
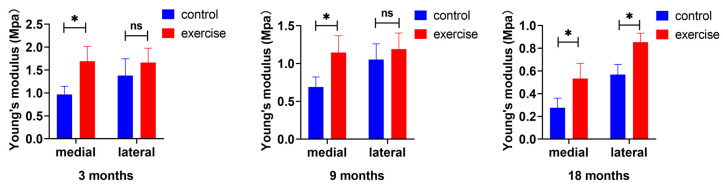
Exercise improves the Young’s modulus of a rat’s knee cartilage. Young’s modulus of the cartilage in the medial and lateral tibial plateaus of rats aged 3 months, 9 months, and 18 months (n = 3 each) in the exercise and control groups. Data are represented as mean ± S.E.M. *p* * < 0.05. ns: no significance.

**Figure 3 diagnostics-13-03193-f003:**
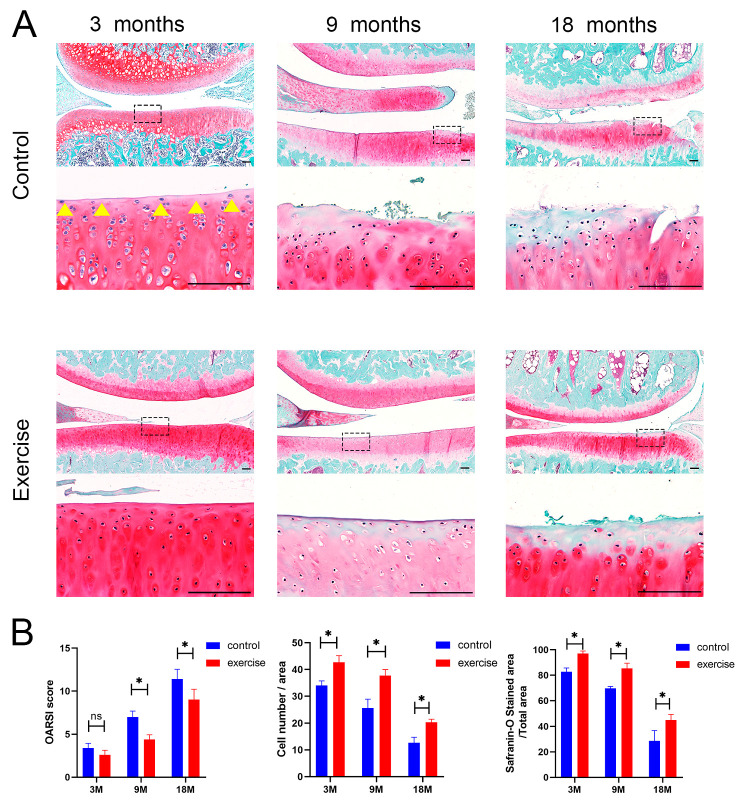
Exercise increases the number of cells and the content of surrounding GAG in the SFZ. Scale bar: 100 μm. (**A**) Safranin O/fast green staining in rats aged 3 months, 9 months, and 18 months (n = 5 each) in the exercise and control groups. (**B**) Histomorphometric analyses for Osteoarthritis Research Society International (OARSI) scores, cell number/area, and Safranin O-stained area/total area in the superficial zone (yellow arrow) in rats aged 3 months, 9 months, and 18 months in the exercise and control groups. Data are represented as mean ± S.E.M. *p* * < 0.05. ns: no significance. The dotted squares represent the area selected for magnification.

**Figure 4 diagnostics-13-03193-f004:**
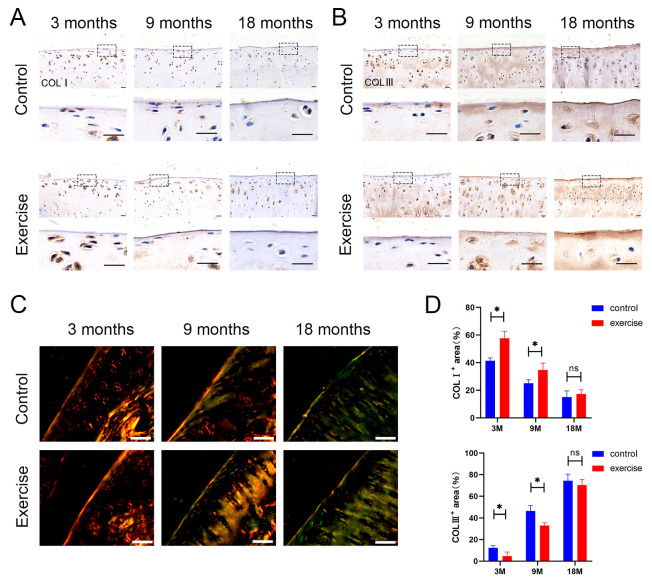
Exercise changes the content and distribution of type Ⅰ and Ⅲ collagens in the SFZ. Scale bar: 20 μm. Immunostaining for (**A**) type Ⅰ and (**B**) type Ⅲ collagens in knee sections of rats aged 3 months, 9 months, and 18 months (n = 5 each) in the exercise and control groups. (**C**) Sirius Red staining of the superficial zone of cartilage. Type I collagen fibers appear intense orange-yellow or bright red, while type Ⅲ collagen fibers appear green. (**D**) Immunohistochemical analyses for positive area percentages of type Ⅰ and type Ⅲ collagens in the superficial zone. Data are represented as mean ± S.E.M. *p* * < 0.05. ns: no significance. The dotted squares represent the area selected for magnification.

**Figure 5 diagnostics-13-03193-f005:**
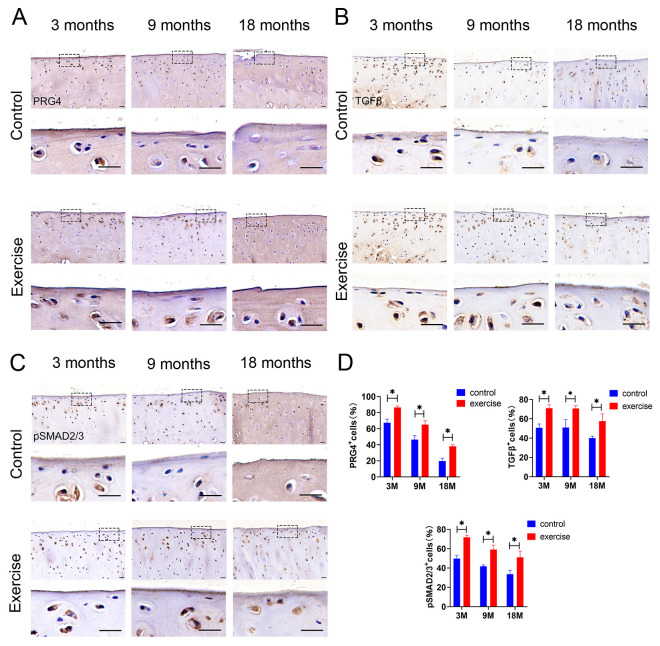
Exercise increases the protein expression of PRG4, TGFβ, and pSmad2/3. Scale bar: 20 μm. Immunostaining for (**A**) PRG4, (**B**) TGFβ, and (**C**) pSmad2/3 on knee sections of rats aged 3 months, 9 months, and 18 months (n = 5 each) in the exercise and control groups. (**D**) Immunohistochemical analyses for positive cell percentages of PRG4, TGFβ, and pSmad2/3 in the superficial zone (scale bar = 100 μm). Data are represented as mean ± S.E.M. *p* * < 0.05. The dotted squares represent the area selected for magnification.

**Figure 6 diagnostics-13-03193-f006:**
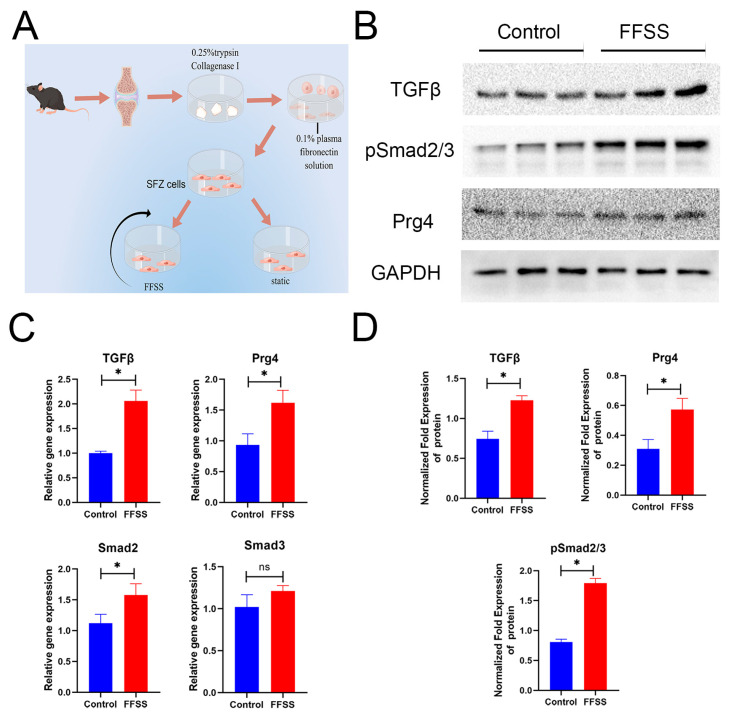
Shear force interferes with SFZ cells and promotes the expression of TGFβ, pSmad2/3, and PRG4. (**A**) Procedure for extraction, isolation, and shear force intervention of SFZ cells. (**B**) Western blotting for protein expression of TGFβ, pSmad2/3, and PRG4 in the FFSS and control groups. (**C**) RT-qPCR analyses for gene expression of TGFβ, Smad2, Smad3, and PRG4 in the FFSS and control groups. (**D**) Western blotting statistical analyses for these key indicators. Data are represented as mean ± S.E.M. *p* * < 0.05. ns: no significance.

**Table 1 diagnostics-13-03193-t001:** Primer sequences used for real-time PCR assay.

Primers	Sequences (5’ to 3’)
Prg4	CACCATCTCCACCACGCAGAAT
	TGCTGAATGTTGCCACCTCTCTTG
Erg	GGTCTTGAAGGTCCCGATGC
	CACTCTGCGCTCATTTGTGG
TenascinC	CAGTACCACGGCTACCACAG
	CATTCTCCGATGCCGTCCAG
TGFβ	ATGGTGGACCGCAACAACGC
	GGCACTGCTTCCCGAATGTCTG
Smad2	CCGTGCTCCCTCCGTCTTCC
	CTGCCGCCCGCTGATTGG
Smad3	TTGACAGAGAGCAACACAGTAT
	CTTCATCCAGATCGATTGCTTG
GAPDH	GGTCCCAGCTTAGGTTCATCA
	ATCCGTTCACACCGACCTTC

## Data Availability

The data sets are not publicly available due to restrictions under the license for the current study. However, they are available on reasonable request from the corresponding author.

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
