# Peer review of "Effect of Moderate Exercise on the Superficial Zone of Articular Cartilage in Age-Related Osteoarthritis"

_diagnostics, 2023, doi:10.3390/diagnostics13203193_

Round 1

Reviewer 1 Report

I congratulate for the prepared manuscript and the conducted research! The authors’ aim was to assess the possible effects of exercise on the superficial zone of osteoarticular cartilage in the knee joint of rats.  

Abstract 

Well written, the followings shall be considered: 

Please revise the need for abbreviation in Line 20: “... 18-month-old SD rats were...” 

Please revise the sentence in Line 23: “... X-ray and micro-CT and were quantitated...” 

1.Introduction 

The importance of the study is well supported by the referenced literature and this section is well written. 

The authors stating (including Abstract): 

Line19-20: “...This study aimed to evaluate the effect of exercise on the superficial zone of the osteoarticular cartilage during osteoarthritis progression...”  

Line 69-70: “...we aimed to explore changes in the content and distribution of collagen and Prg4 in the superficial zone during exercise....” 

Line 83-85: “...In the present study, we used moderate exercise methods to explore the effects of exercise on the cartilage and cartilage surfaces of rats of different ages....” 

The phrasing of aim(s) shall be clearly clarified instead of instead of stated in slightly different ways. 

2. Materials and Methods 

The methods are written, but the sample sizes which were used at the different methods must be clearly provided (micro-CT, Nanoindentation measurements, Histological staining and immunohistochemistry, etc.)  

Additional remarks in this section:  

Please revise Line 93: “... temperature was 20 ± 3oC, and the relative…" 

In X-ray and micro-computed tomography (μCT) analysis section providing information of the flow of the plain radiography assessment is missing (e.g. how many observers were involved). Please provide more detail! 

Please revise the need for capitalization in Line 110:  “... were as follows: 70 KV...”  

Instead of “...layer thickness, 31.2 microns;…" in Line 110-111 I recommend providing the voxel size as well as the filter, which was used during the image acquisition. 

Please provide the name of software with affiliation by which the 3D analyses were performed. 

Please revise Line 142-143:  “... 37oC with 10% goat...”; “... at 4oC with...” 

Please provide the affiliation in Line 149: “... Image-Pro Plus 6.0 software...” 

3. Results 

Inproper location of the sentence, which stating conclusion prior to describing the results (Line 197-199): “...Exercise has a significant influence on the degeneration of the superficial zone of the cartilage and is an effective way to treat and relieve the early symptoms of OA...” 

4. Discussion 

In the Discussion chapter, the author summarizes and compares the results of the conducted examinations with the current literature on the investigated topic. 

My remarks: 

Please revise the need for capitalization in Line 319: “... for two reasons: Firstly, the main...” 

5. Conclusions 

I suggest the revision of conclusion and I recommend a more moderate wording (Line 417-418): “...significant role in delaying the degeneration of the superficial zone of the cartilage...” Namely, the type of cartilage shall be specified to which the statement is valid. 

Formatting of References is appropriate. The resolution of Figure S2 needs to be improved. 

As a summary I found the manuscript to be technically sound and the claims are convincing, though minor changes shall be performed. I recommend the manuscript for minor revision.

Author Response

Response to Reviewer 1 Comments

1. Summary

Thank you very much for taking the time to review this manuscript. Please find the detailed responses below and the corrections highlighted in the re-submitted files.

2. Point-by-point response to Comments and Suggestions for Authors

Comments 1: Abstract

Well written, the followings shall be considered:

â‘   Please revise the need for abbreviation in Line 20: “... 18-month-old SD rats were...”.

â‘¡  Please revise the sentence in Line 23: “... X-ray and micro-CT and were quantitated...”.

Response 1: Thank you for pointing these out. We agree with these comments. Therefore, we have revised point-by-point:

â‘   We revised “... 18-month-old SD rats were...”to “... 18-month-old Sprague-Dawley rats were...”, this change can be found – page1, and line21.

â‘¡  We revised “... X-ray and micro-CT and were quantitated...”to “...X-ray and micro-CT were quantitated...”, this change can be found – page1, and line23.

Comments 2: 1. Introduction

The authors stating (including Abstract):

Line19-20: “...This study aimed to evaluate the effect of exercise on the superficial zone of the osteoarticular cartilage during osteoarthritis progression...” 

Line 69-70: “...we aimed to explore changes in the content and distribution of collagen and Prg4 in the superficial zone during exercise....”

Line 83-85: “...In the present study, we used moderate exercise methods to explore the effects of exercise on the cartilage and cartilage surfaces of rats of different ages....”

The phrasing of aim(s) shall be clearly clarified instead of instead of stated in slightly different ways.

Response 2: Agree. Therefore, we have clearly clarified the phrasing of aim(s). this change can be found – page 2, and line70,84-86.

Comments 3: 2. Materials and Methods

â‘    The methods are written, but the sample sizes which were used at the different methods must be clearly provided (micro-CT, Nanoindentation measurements, Histological staining and immunohistochemistry, etc.) 

Additional remarks in this section: 

â‘¡   Please revise Line 93: “... temperature was 20 ± 3oC, and the relative…"

â‘¢ In X-ray and micro-computed tomography (μCT) analysis section providing information of the flow of the plain radiography assessment is missing (e.g. how many observers were involved). Please provide more detail!

â‘£ Please revise the need for capitalization in Line 110: “... were as follows: 70 KV...” 

⑤ Instead of “...layer thickness, 31.2 microns;…" in Line 110-111 I recommend providing the voxel size as well as the filter, which was used during the image acquisition.

â‘¥ Please provide the name of software with affiliation by which the 3D analyses were performed.

⑦ Please revise Line 142-143: “... 37oC with 10% goat...”; “... at 4oC with...”

⑧ Please provide the affiliation in Line 149: “... Image-Pro Plus 6.0 software...” 

Response 3: Agree. We have, accordingly, revised point-by-point.

â‘   The sample sizes which were used at the different methods were clearly provided (micro-CT this change can be found – page3, and line 105, Nanoindentation measurements this change can be found – page3, and line 127, Histological staining and immunohistochemistry this change can be found – page4, and line 142).

â‘¡  We revised “... temperature was 20 ± 3oC, and the relative…"to “... temperature was 20 ± 3℃…". this change can be found – page3, and line 94.

â‘¢  We provide more detail of the flow of the plain radiography assessment. this change can be found – page3, and line 110-111,124-125.

â‘£  We revised “... were as follows: 70 KV...”to “... were as follows: 70 kV...”. this change can be found – page3, and line 114.

⑤  We provided the voxel size as well as the filter instead of “...layer thickness, 31.2 microns;…". this change can be found – page3, and line 114.

â‘¥  We provide the name of 3D analyses software with affiliation. this change can be found – page3, and line 124.

⑦  We revised “... 37oC with 10% goat...”; “... at 4oC with...” to“... 37℃ with 10% goat...”; “... at 4℃ with...”. this change can be found – page4, and line 153,154.

⑧  We provide the affiliation of Image-Pro Plus 6.0 software. this change can be found – page4, and line 160.

Comments 4: Results

Inproper location of the sentence, which stating conclusion prior to describing the results (Line 197-199): “...Exercise has a significant influence on the degeneration of the superficial zone of the cartilage and is an effective way to treat and relieve the early symptoms of OA...”

Response 4: Agree. Therefore, the inappropriate comments have been deleted, we presented these results in our conclusions. this change can be found – page5, and line 207-210.

Comments 5: In the Discussion chapter, the author summarizes and compares the results of the conducted examinations with the current literature on the investigated topic.

My remarks:

Please revise the need for capitalization in Line 319: “... for two reasons: Firstly, the main...”

Response 5: Agree. We revised “... for two reasons: Firstly, the main...”to “... for two reasons: firstly, the main...”. this change can be found – page12, and line 328.

Comments 6: I suggest the revision of conclusion and I recommend a more moderate wording (Line 417-418): “...significant role in delaying the degeneration of the superficial zone of the cartilage...” Namely, the type of cartilage shall be specified to which the statement is valid.

Response 6: Agree. We specified the type of cartilage as“... significant role in delaying the degeneration of the superficial zone of the knee articular cartilage during age-related osteoarthritis progression...”. this change can be found – page14, and line 439.

Comments 7: The resolution of Figure S2 needs to be improved.

Response 7: The resolution of Figure S2 has been improved. this change can be found – page15, and line 448.

Reviewer 2 Report

Dear authors,

I would like to thank you for this detailed article. It is extremely important for a clinician to understand the pathophysiology and the progression of osteoarthritis and to address it therapeutically. The structure of the paper is very structured and profound.

I'm interested in how the density measurement was carried out with a micro CT when a 3D standard microstructure analysis was carried out. Is it a surface measurement? Then how was the volume calculated?

To ensure that the results were meaningful, young test animals were used because the superficial zone of articular cartilage in older rats is completely worn out. In reality, the clinic shows that it is precisely older patients who suffer from osteoarthritis. This would mean that the results presented here cannot be implemented in everyday clinical practice.

I hope that these points can be discussed again.

Author Response

Response to Reviewer 2 Comments

1. Summary

Thank you very much for taking the time to review this manuscript. Please find the detailed responses below and the corrections highlighted in the re-submitted files.

2. Point-by-point response to Comments and Suggestions for Authors

Comments 1: I'm interested in how the density measurement was carried out with a micro CT when a 3D standard microstructure analysis was carried out. Is it a surface measurement? Then how was the volume calculated?

Response 1: Thank you for pointing this out. we provide a specific volumetric calculation method and density measurement. It is not a surface measurement. The volume of interest (VOI) consisted of a stack of ROIs drawn over 36 cross-sections, resulting in a height of 0.56 mm. The VOI included the subchondral trabecular bone starting below the subchondral plate, and extending distally towards the growth plate, excluding both the cortical bone and growth plate interface. The VOI used was of the same size and shape. this change can be found – page 3, and line116-120.

Comments 2: In reality, the clinic shows that it is precisely older patients who suffer from osteoarthritis. This would mean that the results presented here cannot be implemented in everyday clinical practice. I hope that these points can be discussed again.

Response 2: Agree. At present, the clinic shows that it is older patients who suffer from osteoarthritis. What may explain this result is that most of the research on the pathogenesis and treatment of cartilage degeneration mainly focuses on the overall process of OA, ignoring the research on the early stage of cartilage degeneration. Most patients are not diagnosed with OA until they develop relevant symptoms. And the reason why we focused on the superficial zone of cartilage is we found that in the very early stages of degeneration, there was no change in the middle and deep zone of the articular cartilage while the superficial zone of the articular cartilage was significantly changed in the histological diagnosis. Our findings that moderate exercise intervention in middle and old age can delay the degeneration of the articular cartilage by protecting the environment of the superficial cells and cell matrix of the articular cartilage may provide the new idea for the prevention of early cartilage degeneration at a relatively young age rather than at old age. These discussions can be found– page12, and line332-344.